# Exploring Vulnerabilities of BERT-based APIs

## Abstract

Natural language processing (NLP) tasks, ranging from text classification to text generation, have been revolutionised by pretrained BERT models. This allows corporations to easily build powerful APIs by encapsulating fine-tuned BERT models. These BERT-based APIs are often designed to not only provide reliable service but also protect intellectual properties or privacy-sensitive information of the training data. However, a series of privacy and robustness issues may still exist when a fine-tuned BERT model is deployed as a service. In this work, we first present an effective model extraction attack, where the adversary can practically steal a BERT-based API (the target/victim model). We then demonstrate: (1) how the extracted model can be further exploited to develop effective attribute inference attack to expose sensitive information of the training data of the victim model; (2) how the extracted model can lead to highly transferable adversarial attacks against the victim model. Extensive experiments on multiple benchmark datasets under various realistic settings validate the potential privacy and adversarial vulnerabilities of BERT-based APIs.

## 1 Introduction

The emergence of **B**idirectional **E**ncoder **R**epresentations from **T**ransformers (BERT) (Devlin et al., 2018) has revolutionised the natural language processing (NLP) field, leading to state-of-the-art performance on a wide range of NLP tasks with minimal task-specific supervision. In the meantime, with the increasing success of contextualised pretrained representations for transfer learning, powerful NLP models can be easily built by fine-tuning the pretrained models like BERT or XLNet (Yang et al., 2019). Building NLP models on pretrained representations typically only require several task-specific layers or just a single feedforward layer on top of BERT. To protect data privacy, system integrity and Intellectual Property (IP), commercial NLP models such as task-specific BERT models are often made indirectly accessible through pay-per-query prediction APIs (Krishna et al., 2019) . This leaves model prediction the only information an attacker can access.

Prior works have found that existing NLP APIs are still vulnerable to model extraction attack, which reconstructs a copy of the remote NLP model based on carefully-designed queries and the outputs of the API (Krishna et al., 2019; Wallace et al., 2020). Pretrained BERT models further make it easier to apply model extraction attack to specialised NLP models obtained by fine-tuning pretrained BERT models (Krishna et al., 2019). In addition to model extraction, it is important to ask the following two questions: 1) will the extracted model also leaks sensitive information about the training data in the target model; and 2) whether the extracted model can cause more vulnerabilities of the target model (i.e. the black-box API).

To answer the above two questions, in this work, we first launch a model extraction attack, where the adversary queries the target model with the goal to steal it and turn it into a white-box model. With the extracted model, we further demonstrate that: 1) it is possible to infer sensitive information about the training data; and 2) the extracted model can be exploited to generate highly transferable adversarial attacks against the remote victim model behind the API. Our results highlight the risks of publicly-hosted NLP APIs being stolen and attacked if they are trained by fine-tuning BERT.

**Contributions:** First, we demonstrate that the extracted model can be exploited by an attribute inference attack to expose sensitive information about the original training data, leading to a significant privacy leakage. Second, we show that adversarial examples crafted on the extracted model are highly

transferable to the target model, exposing more adversarial vulnerabilities of the target model. Third, extensive experiments with the extracted model on benchmark NLP datasets highlight the potential privacy issues and adversarial vulnerabilities of BERT-based APIs. We also show that both attacks developed on the extracted model can evade the investigated defence strategies.

## 2 RELATED WORK

### 2.1 MODEL EXTRACTION ATTACK (MEA)

Model extraction attacks (also referred to as "stealing" or "reverse-engineering") have been studied both empirically and theoretically, for simple classification tasks (Tramèr et al., 2016), vision tasks (Orekondy et al., 2019), and NLP tasks (Krishna et al., 2019; Wallace et al., 2020). As opposed to stealing parameters (Tramèr et al., 2016), hyperparameters (Wang & Gong, 2018), architectures (Oh et al., 2019), training data information (Shokri et al., 2017) and decision boundaries (Tramèr et al., 2016; Papernot et al., 2017), in this work, we attempt to create a local copy or steal the functionality of a black-box victim model (Krishna et al., 2019; Orekondy et al., 2019), that is a model that replicates the performance of the victim model as closely as possible. If reconstruction is successful, the attacker has effectively stolen the intellectual property.

Furthermore, this extracted model could be used as a reconnaissance step to facilitate later attacks (Krishna et al., 2019). For instance, the adversary could use the extracted model to facilitate private information inference about the training data of the victim model, or to construct adversarial examples that will force the victim model to make incorrect predictions.

### 2.2 ATTRIBUTE INFERENCE ATTACK

Fredrikson et al. (2014) first proposed *model inversion attack* on biomedical data. The goal is to infer some missing attributes of an input feature vector based on the interaction with a trained ML model. Since deep neural networks have the ability to memorise arbitrary information (Zhang et al., 2017), the private information can be memorised by BERT as well, which poses a threat to information leakage (Krishna et al., 2019). In NLP application, the input text often provides sufficient clues to portray the author, such as gender, age, and other important attributes. For example, sentiment analysis tasks often have privacy implications for authors whose text is used to train models. Prior works (Coavoux et al., 2018) have shown that user attributes can be easily detectable from online review data, as used extensively in sentiment analysis results (Hovy et al., 2015). One might argue that sensitive information like gender, age, location and password are all not explicitly included in model predictions. Nonetheless, model predictions are produced from the input text, it can meanwhile encode personal information which might be exploited for adversarial usages, especially a modern deep learning model owns more capacity than they need to perform well on their tasks (Zhang et al., 2017). The naive solution of removing protected attributes is insufficient: other features may be highly correlated with, and thus predictive of, the protected attributes (Pedreshi et al., 2008).

### 2.3 ADVERSARIAL TRANSFERABILITY AGAINST NLP SYSTEM

An important property of adversarial examples is their transferability (Szegedy et al., 2014; Goodfellow et al., 2015; Papernot et al., 2017). It has been shown that adversarial examples generated against one network can also successfully fool other networks (Liu et al., 2016; Papernot et al., 2017), especially the adversarial image examples in computer vision. Similarly, in NLP domain, adversarial examples that are designed to manipulate the substitute model can also be misclassified by the target model are considered transferable (Papernot et al., 2017; Ebrahimi et al., 2018b). Adversarial transferability against NLP system remains largely unexplored. Few recent works have attempted to transfer adversarial examples to the NLP systems (Sun et al., 2020; Wallace et al., 2020), however, it is oblivious how the transferability works against BERT-based APIs, and whether the transferability would succeed when the victim model and the substitute (extracted) model have different architectures.

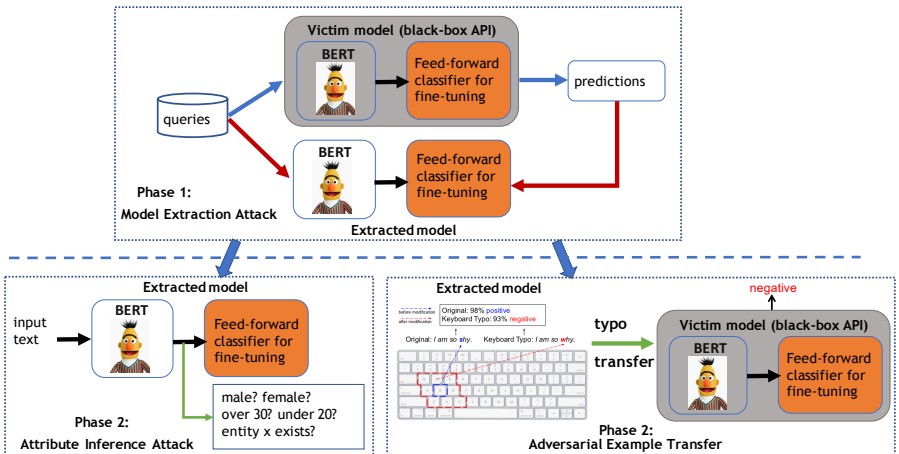

Figure 1: The workflow of the proposed attacks. Phase 1 (model extraction attack): we first sample queries, label them using the victim API, and then train an extracted model on the resulting data. Phase 2: we mount attribute inference attack (bottom left) and transfer adversarial attack (bottom right) based on the extracted model. For attribute inference attack, we train an extra attack model to infer the demographic attributes from BERT representation on any input text, causing privacy leakage of the sensitive attributes of given text. For transferred adversarial attack, we first generate adversarial typo examples on the extracted model, then apply them to attack the victim API.

## 3   ATTACKING BERT-BASED API

In this work, we consider an adversary attempting to steal or attack BERT-based APIs, either for financial gain or to exploit private information or model errors. As shown in Figure 1, the whole attack pipeline against BERT-based APIs can be summarised into two phases. In phase one (*model extraction attack* (MEA)), we first sample queries, label them by the victim API, and then train an extracted model on the resulting data. In phase two, we conduct *attribute inference attack* (AIA) and *adversarial example transfer* (AET) based on the extracted model. We empirically validate that the extracted model can help *enhance* privacy leakage and adversarial example transferability in Section 4.3 and Section 4.4.

We remark that our attack pipeline is applicable to many remote BERT-based APIs, as we assume: (a) the capabilities required are limited to observing model output by the APIs; (b) the number of queries is limited.

### 3.1   VICTIM MODEL: BERT-BASED API

Modern NLP systems are typically based on a pretrained BERT (Devlin et al., 2018; Liu et al., 2019a; Nogueira & Cho, 2019; Joshi et al., 2020). BERT produces rich natural language representations which transfer well to most downstream NLP tasks (sentiment analysis, topic classification, etc.). Modern NLP systems typically leverage the fine-tuning methodology by adding a few task-specific layers on top of the publicly available BERT base,[1] and fine-tune the whole model.

### 3.2   MODEL EXTRACTION ATTACK (MEA)

Model extraction attack aims to steal an intellectual model from cloud services (Tramèr et al., 2016; Orekondy et al., 2019; Krishna et al., 2019; Wallace et al., 2020). In this attack, we assume the victim model is a commercially available black-box API. An adversary with black-box query access to the victim model attempts to reconstruct a local copy ("extracted model") of the victim model. In a nutshell, we perform model extraction attack in a transfer learning setting, where both the adversary and the victim model fine-tune a pretrained BERT. The goal is to extract a model with comparable accuracy to the victim model. Generally, MEA can be formulated as a two-step approach, as illustrated by the top figure in Figure 1:

---

[1]https://github.com/google-research/bert

| Dataset | Private Variable | #Train | #Dev | #Test | Category |
|---------|------------------|--------|------|-------|----------|
| TP-US | age, gender | 22,142 | 2,767 | 2,767 | sentiment analysis |
| AG | entity | 11,657 | 1,457 | 1,457 | topic classification |
| AG full | - | 112k | 1,457 | 1,457 | topic classification |
| Blog | age, gender | 7,098 | 887 | 887 | topic classification |
| Yelp | - | 520k | 40,000 | 1,000 | sentiment analysis |

Table 1: Summary of NLP datasets.

1. Attacker crafts a set of inputs as queries (transfer set), then sends them to the victim model (BERT-based API) to obtain predictions;

2. Attacker reconstructs a copy of the victim model as an "extracted model" by using the queried query-prediction pairs.

Since the attacker does not have training data for the target model, we apply a task-specific query generator to construct $m$ queries $\{x_i\}_1^m$ to the victim model. For each $x_i$, target model returns a $K$-dim posterior probability vector $y_i \in [0,1]^k$, $\sum_k y_i^k = 1$. The resulting dataset $\{x_i, y_i\}_1^m$ is used to train the extracted model. Once the extracted model is obtained, the attacker does not have to pay the provider of the original API anymore for the prediction of new data points.

### 3.3 ATTRIBUTE INFERENCE ATTACK (AIA)

Next, we investigate how to use the extracted model to aid the attribute inference of the private training data of the victim model, *i.e., attribute inference attack* (AIA) (Song & Raghunathan, 2020). We remark that AIA is different from inferring attribute distribution as in model inversion attack (Yeom et al., 2018). The intuition behind AIA is that the BERT representation generated by the extracted model can be used to infer the sensitive attribute of the private training data of the victim model (Li et al., 2018b; Coavoux et al., 2018; Lyu et al., 2020b). Note that in our work, the only explicit information that is accessible to the attacker is model prediction given by the victim model to the chosen inputs, rather than the original BERT representation. We specifically exploit BERT representation of the extracted model, as it encodes the most informative message for the follow-up classification. A more detailed description can be referred to Appendix B.

### 3.4 ADVERSARIAL EXAMPLE TRANSFER (AET)

Due to the success of BERT-based models, numerous works have been proposed to evaluate the vulnerability of BERT based models to adversarial attacks (Jin et al., 2019; Sun et al., 2020). However, most recent works for adversarial example transfer focus on the black-box setting (Gao et al., 2018; Ebrahimi et al., 2018a). In such a setting, the adversary attacks the model via the query feedback only. To circumvent this issue, we leverage the transferability of adversarial examples: we first generate adversarial examples for our extracted model, then transfer them to the BERT-based APIs. The intuition lies in two facts: 1) the rationale of a good model should rely on the salient words; 2) the functionally similarity between our extracted model and the victim model allows for the direct transfer of adversarial examples obtained via gradient-based attacks, which is able to locate the most informative words (Sun et al., 2020). Here our extracted model serves as a surrogate to craft adversarial examples in a white-box manner.

## 4 EXPERIMENTS AND ANALYSIS

### 4.1 NLP TASKS AND DATASETS

We extract models on four diverse NLP datasets that focus on two main tasks: sentiment analysis and topic classification. The four NLP datasets include TP-US from Trustpilot Sentiment dataset (Hovy et al., 2015), AG news corpus (Del Corso et al., 2005), Blog posts dataset from the blog authorship corpus (Schler et al., 2006), and YELP dataset (Zhang et al., 2015). Table 1 summarises the statistics of the used datasets. A more detailed description can be referred to Appendix A.

| Model | Query Size | AG news | AG news (full) | Blog | TP-US | Yelp |
|---|---|---|---|---|---|---|
| Victim model / API | | 79.99 | 94.47 | 97.07 | 85.53 | 95.57 |
| All Same | | **80.83** | **94.54** | **96.77** | **86.48** | **95.72** |
| Data Different (review) | 1x | 69.94 | 88.63 | 88.16 | 85.15 | 94.06 |
| | 5x | 75.29 | 91.27 | 92.75 | 85.82 | 94.95 |
| Data Different (news) | 1x | 71.95 | 90.47 | 83.13 | 84.15 | 91.06 |
| | 5x | 75.82 | 92.26 | 87.64 | 85.46 | 93.13 |

Table 2: Accuracy [%] of the victim models and the extracted models among different datasets in terms of domains and sizes.

## 4.2 MEA

To assess the functional similarity between the victim model and the extracted one, we compare the accuracy of two models, *i.e.,* the closer accuracy indicates a higher similarity. In line with prior work (Krishna et al., 2019), we first choose the size of the resulting transfer set (queries) to be comparable (e.g., 1x) to the size of victim's training set, then scale up to 5x.

**Attack Strategies** We first study model extraction through simulated experiments: we train victim models, query them as if they are black-box APIs, and then train the extracted model to mimic the victim model. We assume that the attacker has access to the freely available pretrained BERT model used by the victim model.

**Query Distribution** To investigate how the data distribution of queries ($P_A$) may impact the attack on the victim model trained on data from $P_V$ (*c.f.,* Table 1), we experiment with the following experiments.

1. We use the same architecture, hyperparameters, and the original data as the victim (All Same).

2. We use the same architecture and hyperparameters as the victim, but sample queries from different distribution (Data Different).

The second scenario makes fewer assumptions and is more realistic and challenging, as the attacker may not know the target data distribution as a prior. Therefore, in addition to the same data distribution as the victim, we additionally investigate the query distribution $P_A$ sourced from the following corpora:

- Reviews data: Yelp and Amazon reviews dataset (Zhang et al., 2015). It is worth noting that we exclude Yelp reviews dataset from the Yelp task to guarantee a fair evaluation.
- News data: CNN/DailyMail dataset (Hermann et al., 2015)

Regarding the experiments of MEA, our general findings from Table 2 include: (1) using same data (All Same) as queries achieves the best extraction performance, validating that the closeness of the domain between the victim training data and queries is positively correlated to the extraction; (2) using same data can achieve comparable accuracies, even outperform the victim models, we hypothesise this is due to the regularising effect of training on soft-labels (Hinton et al., 2015); (3) our MEA is effective despite the fact that queries may come from different distributions. Using samples from different corpora (review and news) as queries, our MEA can still achieve 0.85-0.99× victim models' accuracies when the number of queries varies in {1x,5x}, and the extraction is more successful with 5x queries as expected. This facilitates the follow-up AIA and AET. Even with small query budgets (0.1x and 0.5x), extraction is often successful. More results are available in Appendix C. We also noticed that AG news prefers *news* data, while *reviews* data is superior to *news* data on TP-US, Blog and Yelp. Intuitively, one can attribute this preference to the genre similarity, *i.e., news* data is close to AG news, while distant from TP-US, Blog and Yelp. To rigorously study this phenomenon, we calculate the uni-gram and 5-gram overlapping between test sets and different queries in the 1x setting. Table 3 corroborates that there is a positive correlation between the accuracy and the lexicon similarity. From now, unless otherwise mentioned, because of their effectiveness (*c.f.,*

| Query | AG news (full) | | Blog | | TP-US | | Yelp | |
|---|---|---|---|---|---|---|---|---|
| | uni-gram | 5-gram | uni-gram | 5-gram | uni-gram | 5-gram | uni-gram | 5-gram |
| review | 68.22% | 0.53% | 47.21% | 0.73% | 60.86% | 2.57% | 52.68% | 2.64% |
| news | 72.13% | 1.24% | 44.76% | 0.06% | 51.28% | 0.12% | 38.69% | 0.19% |

Table 3: Percentage of test sets uni-gram and 5-gram overlapping with different queries. Since AG news is derived from AG news (full) and they show a similar distribution, we omit it .

Table 2), we will use *news* data as queries for AG news, and *reviews* data as queries for TP-US, Blog and Yelp.[2]

## 4.3 AIA

For AIA, we conduct our studies on TP-US, AG news and Blog datasets, as there is no matching demographic information for Yelp. AIA is appraised via the following metrics:

- For demographic variables (*i.e.,* gender and age): $1 - X$, where $X$ is the average prediction accuracy of the attack models on these two variables.
- For named entities: $1 - F$, where $F$ is the F1 score between the ground truths and the prediction by the attackers on the presence of all named entities.

Following Coavoux et al. (2018); Lyu et al. (2020a), we denote the value of $1 - X$ or $1 - F$ as *empirical privacy*, *i.e.,* the inverse accuracy or F1 score of the attacker, higher means better empirical privacy, *i.e.,* lower attack performance.

We first randomly split each dataset in Table 1 into two halves. The first half (denoted as $\mathcal{D}_V$) is used to train a victim model, whereas the second half (denoted as $\mathcal{D}_A$) is specifically reserved as the public data for the training of AIA attack model. On the extracted model from MEA, attackers can determine how to infer the private attributes from the BERT representation $\boldsymbol{h}$ of the extracted model over $\mathcal{D}_A$. Each attack model consists of a multi-layer feed forward network and a binary classifier, which takes the $\boldsymbol{h}$ as the inputs and emits the predicted private attribute. Once the attack models are obtained, we measure the empirical privacy by the ability of the attack model to predict accurately the specific private attribute in $\mathcal{D}_V$.

Apart from the standard three corpora used for MEA (*c.f.,* Section 4.2), in AIA, we also consider $\mathcal{D}_A$ (2nd half) as queries, which is derived from the same distribution as $\mathcal{D}_V$. It is worth noting that for AG news, we use the filtered AG news (*c.f.,* Appendix A) with sensitive entity information for AIA.

To gauge the private information leakage, we consider a majority class prediction of each attribute as a baseline. To evaluate whether our extracted model can help enhance AIA, we also take the pretrained BERT without (w/o) fine-tuning as a baseline. Table 4 shows that compared to the pretrained only BERT, the attack model built on the BERT representation of the extracted model indeed largely enhances the attribute inference of the training data of the victim model — more than 4x effective for AG news compared with the majority baseline, even when MEA is based on the queries from different data distribution. This implies that target model predictions inadvertently capture sensitive information about users, such as their gender, age, and other important attributes, apart from the useful

| | AG news | Blog | TP-US |
|---|---|---|---|
| Majority class | 49.94 | 49.57 | 38.15 |
| BERT (w/o fine-tuning) | 69.39 | 44.03 | 49.38 |
| All Same | 22.74 | 36.37 | 36.76 |
| Data Different (2nd half) | | | |
| 1x | 21.01 | 35.98 | 37.34 |
| Data Different (review) | | | |
| 1x | 17.93 | 34.34 | **35.97** |
| 5x | 18.31 | 34.45 | 36.82 |
| Data Different (news) | | | |
| 1x | **15.76** | **33.88** | 36.92 |
| 5x | 17.91 | 35.39 | 37.68 |

Table 4: AIA attack success over different datasets. Note higher value means better empirical privacy, *i.e.,* lower attack success. (2nd half: $\mathcal{D}_A$ reserved for the training of AIA attack model.)

---

[2]Empirically, we do not have access to the training data of the victim model.

information for the main task (*c.f.,* Table 2). By contrast, BERT (w/o fine-tuning) is a plain model that did not contain any information about the target model training data.

Interestingly, compared with queries from the same distribution, Table 4 shows that queries from different distributions make AIA easier (see the best results corresponding to the lower privacy protections in bold in Table 4). We believe this anti-intuitive phenomenon is caused by the posterior probability, as the posterior probability of the same distribution is sharper than that of the different distribution.[3] This argument can be also confirmed from Section 5, in which we use a temperature coefficient $\tau$ at the softmax layer to control the sharpness of the posterior probability.

We speculate that the effectiveness of AIA is related to the undesired deep model memorisation of the victim model, which can be spread to the extracted model through model prediction, incurring information leakage.

We further investigate which kind of attribute is more vulnerable, *i.e.,* the relationship between attribute distribution (histogram variance) and privacy leakage. We empirically found that, compared with the attribute with higher variance, attribute with lower variance is harder to attack.[4]

### 4.4 AET

Since we have access to the parameters of the locally extracted model, we craft white-box adversarial examples on it and test whether such examples are transferable to the target model. We evaluate sample crafting using the metric of transferability, which refers to the percentage of adversarial examples transferring from the extracted model to the victim model. We use Blog, TP-US, AG news (full) and Yelp for AET.

*How We Generate Natural Adversarial Examples?* Following Sun et al. (2020), we first leverage the gradients of the gold labels w.r.t the embeddings of the input tokens to find the most informative tokens. Then we corrupt the selected tokens with the following six sources of typos: 1) Insertion; 2) Deletion; 3) Swap; 4) Mistype: Mistyping a word though keyboard, such as "oh" → "0h"; 5) Pronounce: Wrongly typing due to the close pronounce of the word, such as "egg" → "agg"; 6) Replace-W: Replace the word by the frequent human behavioural keyboard typo based on the statistics.[5] Note that the above operations are constrained by the character distribution on the keyboard. This approach is denoted as *adv-bert*.

To evaluate whether our extracted model is needed to mount transferable attacks, we also attack it by using black-box adversarial examples. Moreover, following (Sun et al., 2020), we also experiment with a variant of adv-bert, where the target tokens are randomly selected instead of from the maximum gradients, namely *random adv-bet*. Compared with the adversarial examples crafted by black-box and random adv-bert approaches, Table 5 shows that the adversarial examples crafted on our extracted model in a white-box manner make the target model

|  |  | AG news (full) | Yelp | TP-US | Blog |
|---|---|---|---|---|---|
|  |  | transferability | | | |
| black-box | deepwordbug | | | | |
|  | 1x | 25.6 | 13.5 | 18.4 | 52.9 |
|  | 5x | 35.3 | 20.7 | 18.2 | 67.8 |
|  | textbugger | | | | |
|  | 1x | 16.1 | 11.3 | 21.3 | 41.2 |
|  | 5x | 24.7 | 16.3 | 21.1 | 62.7 |
|  | textfooler | | | | |
|  | 1x | 18.5 | 12.3 | 27.5 | 34.7 |
|  | 5x | 24.9 | 16.9 | 27.1 | 64.4 |
| Rand | adv-bert | | | | |
|  | 1x | 32.5 | 13.4 | 33.3 | 56.2 |
|  | 5x | 45.5 | 21.3 | 34.0 | 66.3 |
| w-box (ours) | adv-bert | | | | |
|  | 1x | 47.5 | 17.8 | **48.6** | 64.9 |
|  | 5x | **53.6** | **25.2** | 47.3 | **76.5** |

Table 5: Transferability is the percentage of adversarial examples transferring from the extracted model to the victim model. Higher is better. deepwordbug (Gao et al., 2018); textbugger (Li et al., 2018a); textfooler (Jin et al., 2019); adv-bert (Sun et al., 2020). w-box: white-box attack. Rand: randomly select a word.

---

[3]Please refer to the Appendix C for the detailed analysis.

[4]Please refer to the Appendix C for the detail.

[5]https://en.wikipedia.org/wiki/Wikipedia:Lists_of_common_misspellings

| Victim | Extracted | TP-US | | | AG news (full) | | |
|---|---|---|---|---|---|---|---|
| | | victim ↑ | MEA ↑ | AIA ↓ | victim ↑ | MEA ↑ | AET ↑ |
| BERT-large | BERT-base | 86.82 | 85.36 | 36.65 | 94.75 | 89.88 | 42.7 |
| RoBERTa-large | BERT-base | 87.20 | 85.72 | 37.33 | 95.40 | 89.74 | 27.7 |
| RoBERTa-base | BERT-base | 86.66 | 85.40 | 37.52 | 95.18 | 89.45 | 36.4 |
| XLNET-large | BERT-base | 87.21 | 85.99 | 37.68 | 95.28 | 89.66 | 32.7 |
| XLNET-base | BERT-base | 86.91 | **86.13** | 38.09 | 94.74 | 89.27 | 34.4 |
| BERT-base | BERT-base | 85.53 | 85.15 | **35.97** | 94.47 | **90.47** | **47.5** |

Table 6: Attack performance on TP-US and AG news (full) with mismatched architectures between the victim and the extracted model.

more vulnerable to adversarial examples in terms of transferability — more than twice effective in the best case. This validates that our extracted model, which is designed to be a high-fidelity imitation of the victim model, considerably enhances the adversarial example transferability, thus severely damaging the output integrity of the target model.

We examine potential factors that contribute to the successful transferability. We found that collecting a larger number of queries contributes to a better attack performance, *i.e.,* 5x queries generally results in much better transferability compared with 1x. This implies that the extracted model with higher fidelity (closer to the victim model, *c.f.,* Table 2) can considerably enhance the adversarial example transferability.

### 4.5 ARCHITECTURE MISMATCH

In practice, it is more likely that the adversary does not know the victim's model architecture. A natural question is whether model extraction is still possible even when the extracted models and the victim models have different architectures. To study the influence of the architectural mismatch, we fix the architecture of the extracted model, while varying the victim model from BERT (Devlin et al., 2018), RoBERTa (Liu et al., 2019b) to XLNET (Yang et al., 2019). According to Table 6, when there is an architecture mismatch between the victim model and the extracted model, the efficacy of AIA and AET is alleviated as expected. However, the leakage of the private information is still severe (*c.f.,* the majority class in Table 4). Surprisingly, we observe that for AG news (full), MEA cannot benefit from a more accurate victim, which is different from the findings in Hinton et al. (2015). We conjecture such difference is ascribed to the distribution mismatch between the training data of the victim model and the queries. We will conduct an in-depth study on this in the future.

## 5 DEFENCE

Although we primarily focus on the vulnerabilities of BERT-based APIs in this work, we briefly discuss several counter strategies the victim model may adopt to reduce the informativeness of prediction while minimising the overall drop in API performance (Shokri et al., 2017).

**Hard label only**. The posterior probability usually leaks more information from the victim model, thus victim model can choose to only return the hard label.

**Softening predictions.** A temperature coefficient $\tau$ on softmax layer manipulates the distribution of the posterior probability. A higher $\tau$ leads to smoother probability, whereas a lower one produces a sharper distribution. When $\tau$ is approaching 0, the posterior probability becomes a hard label.

Table 7 indicates that although varying temperature on softmax cannot defend the victim model against MEA, it is an effective defensive approach to AIA when $\tau = 0.5$, *i.e.,* closer to the hard label. Similarly, compared with ND, hard label can help mitigate all attacks to some extent.[6]

However, there is no defence that is effective against all our attacks (*c.f.,* Table 2, Table 4, Table 5), as all these defences preserve the rank of the most confident label. Models can still be effectively stolen

---

[6]We observe the similar behaviours for Yelp and Blog.

|  | TP-US | | | AG news | | AG news (full) | |
|---|---|---|---|---|---|---|---|
|  | MEA↑ | AIA↓ | AET↑ | MEA↑ | AIA↓ | MEA↑ | AET↑ |
| ND | 85.33 | 36.92 | 48.6 | 71.95 | 15.76 | 90.48 | 47.5 |
| SOFTENING ($\tau = 0.5$) | 85.10 | **37.69** | 50.9 | 72.32 | 20.78 | 90.29 | 33.0 |
| SOFTENING ($\tau = 5$) | 85.33 | 33.17 | 58.7 | 72.48 | 11.32 | 90.87 | 53.3 |
| HARD LABEL | **84.60** | 37.62 | **40.2** | **69.11** | **22.47** | **89.98** | **33.0** |

Table 7: Attack performance under different defences and datasets. ND: no defence; $\tau$: temperature on softmax. For MEA and AET, **lower** scores indicate better defences, conversely for AIA. All experiments are conducted with 1x queries.

and exploited using just the hard label or the smoothed predictions returned by the black-box API. This further validates that the adversary only needs to have access to the victim model's hard label, and does not always need to have access to the confidence scores for our attacks to be successful.

## 6 DISCUSSION

Understanding how well our attacks work in various settings is important for defenders to know how vulnerable their systems are. Extensive experiments in this paper indicate that the privacy and robustness of an NLP system depend on the model complexity as well as the task. For example, the privacy leakage of the victim model becomes more serious by inferring from the extracted model for AG news and Blog, while this phenomenon is less obvious for TP-US dataset (*c.f.,* Table 4). In terms of robustness against adversarial example transferability, Blog is more vulnerable (*c.f.,* Table 5).

Adversarial attacks focus more on the study of the robustness of a model. However, under the context of business, we believe adversarial attacks can also be utilised for other purposes. For instance, if a business competitor manages to spot incorrect predictions, they can improve the robustness of their model while launching an advertising campaign against the victim model with these adversarial examples. If a rival company directly leverages black-box adversarial attacks on the victim model, its owner can detect the suspicious querying, which involves intensive similar queries (Jin et al., 2019; Li et al., 2020; Garg & Ramakrishnan, 2020), thereby banning the abnormal usage. Since queries used for our model extraction are genuine instances generated on the Internet, it is unlikely to be suspended by the cloud services. As evidenced in Section 4.4, the victim model is vulnerable to our proposed AET.

Defence against all our investigated attacks in this work is a hard and open problem. An ideal defence should resist against all the possible attacks while striving to have a minimal impact on legitimate users of the model (Orekondy et al., 2019). While current defences are marginally effective, they may fail when adversaries adapt to the defence — sophisticated adversaries might anticipate these defences and develop simple modifications to their attacks to circumvent these defences (Krishna et al., 2019). We hope that this work highlights the need for more research in the development of effective countermeasures to defend against these attacks, or at least to increase the cost of adversaries.

## 7 CONCLUSIONS

This work goes far beyond only model extraction from BERT-based APIs, we also identified that the extracted model can largely enhance the privacy leakage and adversarial example transferability even in difficult scenarios (*e.g.,* limited query budget, queries from different distributions). Extensive experiments based on representative NLP datasets and tasks under various settings demonstrate the effectiveness of our attacks against BERT-based APIs. We hope that our in-depth investigation can provide new insights, and arouse the awareness of the community for building more trustworthy BERT-based API. A number of avenues for further work are attractive. More broadly, we expect to extend our work to more complex NLP tasks, and develop defences that can ensure privacy, robustness, and accuracy simultaneously.

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

APPENDIX

## A    DATASET DESCRIPTION

**Trustpilot (TP)**    Trustpilot Sentiment dataset (Hovy et al., 2015) contains reviews associated with a sentiment score on a five point scale, and each review is associated with 3 attributes: gender, age and location, which are self-reported by users. The original dataset is comprised of reviews from different locations, however in this paper, we only derive TP-US for study. Following Coavoux et al. (2018), we extract examples containing information of both gender and age, and treat them as the private information. We categorise "age" into two groups: "under 34" (U34) and "over 45" (O45).

**AG news**    We use AG news corpus (Del Corso et al., 2005). This task is to predict the topic label of the document, with four different topics in total. Following (Zhang et al., 2015; Jin et al., 2019), we use both "title" and "description" fields as the input document.

We use full AG news dataset for MEA and AET, which we call AG news (full). As AIA requires entity information, we use the corpus filtered by Coavoux et al. (2018)[7], which we call AG news. The resultant AG news merely includes sentences with the five most frequent person entities, and each sentence contains at least one of these named entities. Thus, the attacker aims to identify these five entities as 5 independent binary classification tasks.

**Blog posts (Blog)**    We derive a blog posts dataset (Blog) from the blog authorship corpus presented (Schler et al., 2006). We recycle the corpus preprocessed by Coavoux et al. (2018), which covers 10 different topics. Similar to TP-US, the private variables are comprised of the age and gender of the author. And the age attribute is binned into two categories, "under 20" (U20) and "over 30" (O30).

**Yelp Polarity (Yelp)**    Yelp dataset is a document-level sentiment classification (Zhang et al., 2015). The original dataset is in a five point scale (1-5), while the polarised version assigns negative labels to the rating of 1 and 2 and positive ones to 4 and 5.

## B    AIA ALGORITHM

The main algorithm for *Attribute Inference Attack* (AIA) is shown in Algorithm 1. For each dataset, once the extracted model $g'_V$ is built, we query $g'_V$ with the available public data $D_A$ to collect the BERT representation $h(x_i)$ for each $x_i \in D_A$. For each sensitive attribute $s$, a specific inference model (*c.f.,* Section 4.3) is trained on $\{(h(x_i), s_i)\}$, in order to infer the private attributes of the interest; in our case, they are gender, age and named entities (*c.f.,* Table 1).

In more detail, in Algorithm 1, given $D_A$, we take all the non-sensitive attributes $x_i$ as input, and the sensitive attribute $s_i$ as label to train an AIA attack model. During test time, attacker could feed the non-sensitive attributes of any input into the trained model to infer the sensitive attribute. In the case when the attacker gets the non-sensitive attributes of any training record of the victim model, the attacker can successfully infer its sensitive attributes, thus causing privacy leakage of the victim model training data (*c.f.,* Table 4, we use the non-sensitive attributes of $D_V$ as test data, and demonstrate the sensitive attribute privacy leakage of $D_V$). Note that the non-sensitive attributes of the victim training data could be accessible to any attacker.

## C    ABLATION STUDY

**Query Size**    Due to the budget limit, malicious users cannot issue massive requests. To investigate the attack performance of model extraction under the low-resource setting, we conduct two additional experiments, which only utilise 0.1x and 0.5x size of the training data of the victim models respectively. According to Table 8, although some datasets such as Blog suffer from a drastic drop, the overall performance of the extracted models is comparable to the victim models. In addition, distant domains

---

[7]https://github.com/mcoavoux/pnet/tree/master/datasets.

---

**Algorithm 1** Attribute inference attack

1: **Input:** extracted model $g'_V$, labelled auxiliary data $D_A = (x_i, s_i)$, BERT representation layer $\boldsymbol{h}$, non-sensitive attributes $x^*$
2: Query $g'_V$ with $D_A$ and collect $\{(\boldsymbol{h}(x_i), s_i) | (x_i, s_i) \in D_A\}$.
3: Train an inference model $f$ on $\{(\boldsymbol{h}(x_i), s_i)\}$.
4: Query $g'_V$ with $x^*$ to get the target BERT representation $\boldsymbol{h}(x^*)$
5: **return** $f(\boldsymbol{h}(x^*))$

---

|  | AG news | AG news (full) | Blog | TP-US | Yelp |
|---|---|---|---|---|---|
| Victim model | 79.99 | 94.47 | 97.07 | 85.53 | 95.57 |
| All Same | 80.83 | 94.54 | 96.77 | 86.48 | 95.72 |
| Data Different (review) |  |  |  |  |  |
| 0.1x | 50.90 | 86.57 | 36.83 | 79.95 | 92.39 |
| 0.5x | 68.13 | 87.31 | 84.59 | 84.21 | 93.25 |
| 1x | 69.94 | 88.63 | 88.16 | 85.15 | 94.06 |
| 5x | 75.29 | 91.27 | 92.75 | 85.82 | 94.95 |
| Data Different (news) |  |  |  |  |  |
| 0.1x | 61.70 | 89.13 | 18.04 | 79.20 | 88.24 |
| 0.5x | 70.56 | 89.84 | 32.92 | 84.18 | 89.76 |
| 1x | 71.95 | 90.47 | 83.13 | 84.15 | 91.06 |
| 5x | 75.82 | 92.26 | 87.64 | 85.46 | 93.13 |

Table 8: The accuracy of victim models and extracted models among different datasets in terms of domains and sizes.

exhibit significant degradation, when compared to the close ones. For example, sampling 0.1x-5x queries from news data present a more stable attack performance against the victim model trained on AG news than Blog.

**Impact Factor on AIA** In Section 6, we found that there is a correlation between the success of AIA and temperature $\tau$ on the softmax layer. We conjecture that the causal factor is the sharpness of the posterior probability, *i.e.,* if the model is less confident on its most likely prediction, then AIA is more likely to be successful. This speculation is confirmed by Figure 2, where the higher posterior probability leads to a higher empirical privacy.

Figure 3 and Table 9 indicate that AIA is also affected by the distribution of attributes. Attributes with higher variances cause more information leakage or a lower empirical privacy. For example, for AG-news, entity 2-4 with higher variances result in lower empirical privacy, while entity 0-1 are more resistant to AIA. For TP-US and Blog, as age and gender exhibit similar distribution, AIA performance gap across these two attributes is less obvious, as evidenced by the last two rows in Table 9.

## D    ADVERSARIAL EXAMPLES

We provide several adversarial examples generated by adv-bert (Sun et al., 2020) in Table 10. Note that all these examples cause a misclassification on both extracted models and victim models.

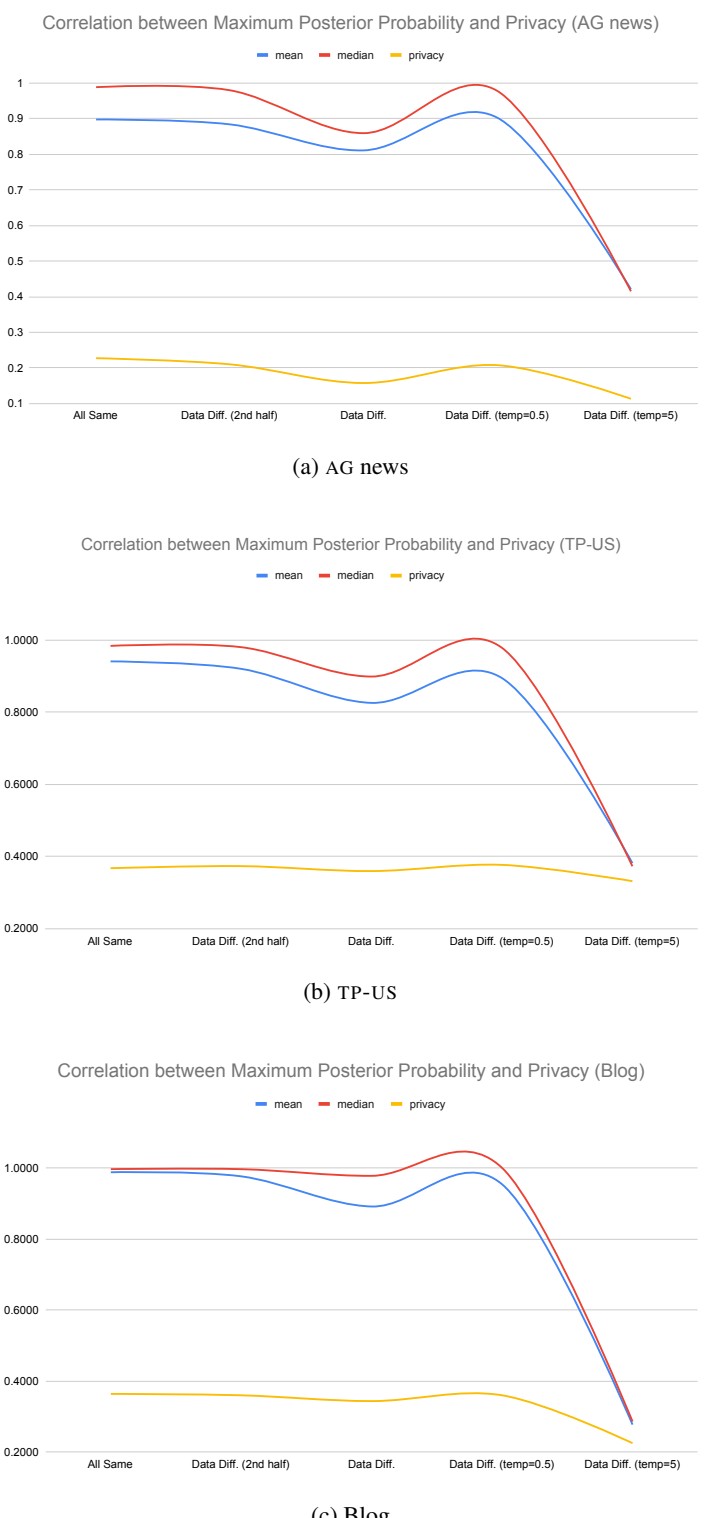

(a) AG news

(b) TP-US

(c) Blog

Figure 2: The correlation between the empirical privacy of AIA and the maximum posterior probability. AG news uses news data as queries, while Blog and TP-US query the victim models with reviews data. **mean** and **median** denote the mean and median of the maximum posterior probability of the queries.

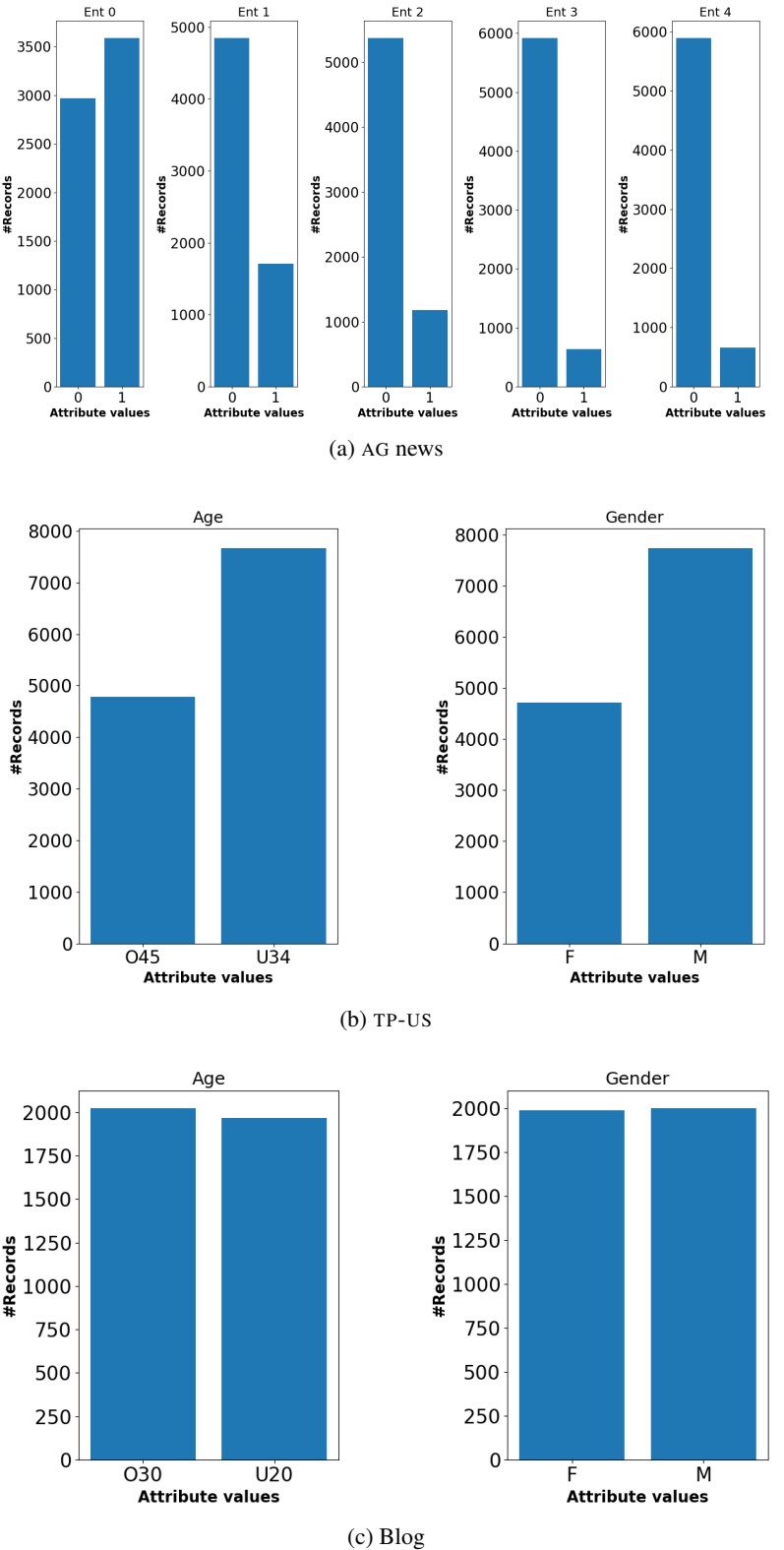

(a) AG news

(b) TP-US

(c) Blog

Figure 3: The distribution of different datasets grouped by private attributes.

| AG news | | | | | |
|---|---|---|---|---|---|
| | entity 0 | entity 1 | entity 2 | entity 3 | entity 4 |
| All Same | 15.61 | 15.10 | 7.71 | 6.95 | 5.49 |
| Data Diff. (news) | 14.79 | 12.38 | 3.84 | 5.33 | 2.02 |

| TP-US | | | Blog | | |
|---|---|---|---|---|---|
| | gender | age | | age | gender |
| All Same | 36.40 | 37.12 | All Same | 32.18 | 39.02 |
| Data Diff. (reviews) | 36.44 | 37.40 | Data Diff. (reviews) | 31.20 | 38.01 |

Table 9: AIA performance on attributes of different datasets. All experiments are conducted with 1x queries.

AG news:

**Original**:
  mps back putin plan for regions . the russian duma backs president putin ' s plan to replace elected regional bosses with his own appointees .

**Adversarial**:
  mps back putin plan for regions . the russian duma backs president putin ' s plan to replace elected regional boss3s with his own appointees .

**Original**:
  baseballer shot on bus . cleveland indians righthander kyle denney was reported to be in a stable condition after being shot in the leg on the team bus yesterday .

**Adversarial**:
  basrballer shot on bus . cleveland indians righthnader kyle denney was reported to be in a stable condition after being shot in the leg on the teqm bus yesterda7

Blog:

**Original**:
  stayed for dinner , and crawled all over him delightfully . it ' s just so

**Adversarial**:
  stayed for dinner , and crawl3d all over him delightfully . it ' s just so

**Original**:
  flowed under the bridge during that time of sharing together , let me

**Adversarial**:
  flowe under the bridge during that time of sharing together , let me

TP-US:

**Original**:
  great company ! excellent service ! fast shipping and never a problem with the order . their products are great and we love the free stuff !

**Adversarial**:
  great company ! excellent service ! fast shipping and nevr a problem with the order . their products are great and we love the free stuff !

**Original**:
  fraud made the purchase unacceptible took longer than anyother key site i have bought from . horrible avoid if you can .

**Adversarial**:
  frsud made the purchase unacceptible took longer than anyother key site i have bought from . borrible avoid if you can .

Yelp:

**Original**:
  sadly really went downhill . they switched to a new vendor , the prices went up and they are no longer having sales .

**Adversarial**:
  safly really went dowhnill . they switched to a new vendor , the prices went up and they are no longer having ssles .

**Original**:
  carbs , carbs , and more carbs . . . . . . this is a horrendous place to eat . steer as far away from this place as you can . minus five if allowed . fries on a crummy beef sandwich .

**Adversarial**:
  carbs , carbs , and more carbs . . . . . . this is a horrednous place to eat . steee as far away from this place as you can . minus five if allowed . fries on a crummy beef sandwich .

Table 10: Adversarial examples generated by adv-bert on different datasets. All of them cause a misclassification. The misspellings are highlighted in red.

