# OpenReview forum: "EXPLORING VULNERABILITIES OF BERT-BASED APIS"
_ICLR.cc/2021/Conference — Reject_

### Official Review · AnonReviewer2 · 2020-10-16
**Interesting results, but I have concerns about experimental setup**

**Rating:** 6
**Confidence:** 5

**Review:**

Summary: This paper is studying the vulnerabilities of modern BERT-based classifiers, which a service provider is hosting using a black-box inference API. Consistent with prior work [2], the authors succeed in extracting high performing copies of the APIs, by training models using the outputs of the API to queries (akin to distillation). The authors then study two attacks on the copy model --- private attribute identification of sentences in the API's training data & adversarial example transfer from the white-box copy model to the black-box API. The authors report high attack success rates, better than those from competitive baselines (which do not require constructing a copy model). A few defences are also explored but are ineffective to prevent these attacks.

-------------------------------------------------

Strengths of the Paper:

1. While model extraction on BERT models has been studied previously [2], this paper goes beyond the setting of utility theft and explores information leakage and adversarial example transfer. These are extremely practical real-world settings. Moreover, the paper uses modern NLP techniques (finetuning BERT), which is ubiquitous in NLP systems these days.

2. The reported attacks seem to significantly outperform some competitive baselines which didn't use an extracted model. While I have concerns about the experimental setup (below), these are very interesting results highlighting vulnerabilities of the models. This can encourage more research in defending against model extraction.

-------------------------------------------------

Weaknesses of the Paper:

1. Query distribution: These distributions seem fairly similar to the downstream task for all datasets, for instance, "reviews" contains Yelp reviews, which is one of the datasets the victim model was trained on (I suspect some amount of overlap at the very least). The best MEA scores are observed when the domains are aligned, which might not be a practical setting for an attacker who has no knowledge of the victim model's training distribution. I suggest, at the very least, authors to provide n-gram overlap statistics between their preferred query distribution and downstream test set (the GPT2 paper [3] had similar statistics). The paper's story will be stronger if a corpus like Wikipedia is used for the query distribution, with the same set of downstream datasets.

2. AIA Attacks: I have a few concerns here. First, isn't access to private attributes in half of the victim data (D_a) too strong an assumption? In a more practical setting, an attacker will have no access to D_a. It's even possible that the attacker doesn't know the output space of attributes. I think the more interesting setting is where the attacker is able to infer some information about the training data without supervising a classifier with gold data (D_a), perhaps using something like model inversion. This information need not be a private binary label, it could even be some canary string like a credit card number [4]. One more concern I had here was regarding the main baseline in this experiment, "BERT (w/o fine-tuning)". I find it quite strange that this is much worse than the majority class in two datasets. What happens when you fine-tune it on D_a? (using the standard practice of [CLS] vector for classification). This is a valid baseline if access to D_a is assumed, I think this will do quite well if it is possible to infer the private variable from the text.

3. Adversarial example transfer: My main concern here is that "transfer rate" by itself is insufficient. You can make transfer rate 100% by retrieving examples from the target adversarial class. The more interesting evaluation is, what fraction of adversarial examples are both (1) transferred correctly; (2) not adversarial to a human (the changes are so minor that humans ignore them). Some kind of human evaluation for (2) will be helpful. Also, a good baseline here would be using adv-bert but with randomly chosen words (instead of white-box gradients), and an upper bound with adv-bert attacks on the victim model itself.

-------------------------------------------------

Overall Recommendation:

While this is a very practically important setting, I'm not entirely convinced the proposed attacks work. My main concerns are regarding some of the experimental decisions and lack of baselines while comparing attacks. Overall I think the paper needs more work to be ready for publication.


-------------------------------------------------

Other Feedback:

While these points are not a make or break for me, they will make the paper stronger. It will be nice to include some fine-grained qualitative analysis of the adversarial examples (along with samples), perhaps highlighting why generating that example would only be possible with access to an extracted model, and confirming the victim API model generates the same example. It will also be nice to see work beyond classification setting. Setups like question answering, machine translation, unconditional text generation are exciting testbeds which might be a lot more vulnerable to AIA style attacks than classifiers. With GPT3, black-box text generation APIs are probably going to get very common in the next 2-3 years!

-------------------------------------------------

Errors / Typos / Stylistic:

I had some trouble understanding parts of the paper. I think with a bit more polishing and careful proof-reading, the paper will be easier to understand. There were also a few incorrect statements. I've pointed them below along with typos / stylistic suggestions,

"commercial NLP models such as Google’s BERT and OpenAI’s GPT-2 (Radford et al., 2019) are often made indirectly accessible through pay-per-query prediction APIs." --> This is not a correct statement, both pretrained models are freely available

"and NLP tasks (Chandrasekaran et al., 2020)." is a mis-citation, you probably wanted to cite Pal et al. 2019 [1] or Krishna et al. 2020 [2] here?

In 3.2 and the Abstract / Intro I would remove the claim that "architecture, hyperparameter is not known", since both the victim / attacker are finetuning BERT.

There's some unnecessary mathiness in 3.2 (variables which are not referred to later on, like f_{bert}_theta*). I would suggest avoiding variables unless you plan to re-use them to reduce confusion.

In Table 3 I would suggest reporting attack success rather than privacy, to be consistent with other tables in paper (higher means more attack success)

Table 4 caption, "Transferability is the ratio" --> "Transferability is the percentage"?

-------------------------------------------------

**After Author Response**: I really appreciate the author's efforts over the course of the rebuttal period for rigorously testing their method with several new baselines in such a short period of time.

For AIA attacks, the baseline numbers provided in the rebuttal are helpful but raise concerns about whether the proposed AIA attacks are working. I find it hard to believe that victim models have less private information than extracted models in 2 out of 3 datasets, and I suspect some other factors are contributing to this counterintuitive trend (like you said, maybe dark knowledge). I will stick with my stance that the AIA setting is broken since you are inferring private attributes using information from an identically distributed D_a (I think model inversion is a more valid setting to measure leakage).

For adversarial attack baselines, I agree with your argument that conducting black-box attacks directly on the victim models may need minimal difference queries which can be detected on the API side. However, you are going to need several orders of magnitude more queries to do extraction in the first place (which may or may not be easy to detect). I still encourage you to run this baseline in the next version of the paper, instead of only doing black-box attacks on extracted models. These minimal difference checks may not be in place, and directly doing black-box attacks on the victim model are much easier than extracting and then constructing adversarial examples. It is good to know what additional benefit you get by doing model extraction.

Overall, I have decided to raise my score to 6 (more like ~5.5-6). This is conditional on the authors performing much more rigorous hypothesis-driven testing in the next version of the paper (just like they did in the rebuttal) to really validate the hypothesis "extracting models make APIs more vulnerable to adversarial attacks".

-------------------------------------------------

References

[1] - https://arxiv.org/abs/1905.09165
[2] - https://arxiv.org/abs/1910.12366
[3] - https://cdn.openai.com/better-language-models/language_models_are_unsupervised_multitask_learners.pdf
[4] - https://arxiv.org/abs/1802.08232

---

> ### Author Response · Authors · 2020-11-19
> **Response to AnonReviewer2**
>
> First of all, We would like to appreciate the reviewer’s suggestion.
>
> [Re: query distribution] Sorry for the miscommunication caused in our paper. We would like to clarify/correct a few key points in the reviewer's comments. First, we removed the Yelp dataset from the queries, when attacking the Yelp task. In addition, when attackers conduct an attack to steal the model, they don’t have access to the in-house training dataset, but they definitely have some prior knowledge of the task as the commercial pay-per-query prediction NLP systems are responsible for publishing a step-by-step document to guide end-users which task the API is used for and how to use their APIs. Thus we selected the datasets relevant to the task of interest.
> Following the reviewer’s suggestion, we also calculate the uni-gram and 5-gram overlapping between test sets and different queries on the 1x setting. According to table3, AG news is close to news data, while TP-US, Blog, and Yelp are similar to reviews data in terms of the lexicons, which corroborates the results of table 2.
>
> [Re: AIA setting]  In terms of AIA, we would like to kindly correct some misunderstandings of the reviewer, with the hope that this will help clarify the experimental setting. We never assume that the attacker can access the private attributes in half of the victim data. In Sec.4.3, we have clearly stated that for each dataset, the first half (denoted as D_V) is used to train a victim model, whereas the second half (denoted as D_A) is specifically reserved as public data for the training of AIA attack model. There is no overlap between D_V and D_A. We also pointed out the difference between our AIA attack and model inversion attack in Sec.3.3. Definitely, model inversion attack can also be conducted upon the extracted model as well, which however is not our main focus in this work.
> In Table 3, we take BERT (w/o fine-tuning) as a baseline as we want to demonstrate that the attack model built on the BERT representation of the extracted model indeed largely enhances the attribute inference of the training data of the victim model. And the reason is quite straightforward: the extracted model trains on the queries and the returned predictions from the victim model, while BERT (w/o fine-tuning) is a plain model that did not contain any information about the target model training data. Hence, it’s not surprising that BERT (w/o fine-tuning) may even get worse performance than the majority class on AG new and TP-US. We have added the extra explanation in our revision in Sec.4.3.
>
> [Re: Adversarial example transfer]
> The first question is about the transfer rate. Many previous works have studied and provided results about the transfer rate between different models [1][2][3]. From these studies, we can see that it is hard to achieve a 100% transfer rate due to the general modification budget.
> The second question is about human evaluation and randomly chosen words. This is also studied by previous studies [1]. Previous results show the randomly chosen word would decrease the attack rate with the limited modification budget. We think we have provided enough results to demonstrate the effectiveness of our attack strategies. However, following your suggestions, we did additional experiments on a variant of adv-bert, where the tokens are selected randomly on table 5 (c.f. the revised version). In this case, our white-box setting still demonstrates superiority. We will include a human evaluation in the final revision as what previous works have done [1][3]
>
> [Other Feedback] We want to correct the point in “victim API model generates the same example”. There is no reason and motivation for the victim API model to generate the adversarial examples, as victim API is the service provider who needs to ensure utility for benign users. In the black-box attacks against the victim API, adversarial examples are generated by the attacker with the aim of compromising the integrity of the victim API [Papernot et al., 2017]. We conduct MEA first in order to steal the victim API, then based on the extracted model, we can generate natural adversarial examples [1] with high transferability to the victim API. Hope this clarifies.
> In terms of other tasks, we agree question answering, machine translation, unconditional text generation are also exciting testbeds. However, we want to emphasize that our work is a pioneering work validating that the subsequent attacks can be done after the model extraction, hence we only investigated several representative classification tasks. It should be easy to extend to other tasks with a minor alteration, which is our concurrent work in another paper.

---

> > ### Author Response · Authors · 2020-11-19
> > **References**
> >
> > [1] Sun, Lichao, et al. "Adv-BERT: BERT is not robust on misspellings! Generating nature adversarial samples on BERT." arXiv preprint arXiv:2003.04985 (2020).
> >
> > [2] Wallace, Eric, Mitchell Stern, and Dawn Song. "Imitation Attacks and Defenses for Black-box Machine Translation Systems." arXiv preprint arXiv:2004.15015 (2020).
> >
> > [3] Li, Jinfeng, et al. "Textbugger: Generating adversarial text against real-world applications." arXiv preprint arXiv:1812.05271 (2018).
> >
> > [4] Song, C. and Shmatikov, V., 2019. Overlearning reveals sensitive attributes. arXiv preprint arXiv:1905.11742.
> >
> > [5] Nicolas Papernot, Patrick McDaniel, Ian Goodfellow, Somesh Jha, Z Berkay Celik, and AnanthramSwami. Practical black-box attacks against machine learning. InProceedings of the 2017 ACM onAsia conference on computer and communications security, pp. 506–519, 2017.

---

> > ### Comment · AnonReviewer2 · 2020-11-20
> > **Thanks for the clarifications, a few questions**
> >
> > Thanks for the detailed response. Here's what I think (happy to discuss more),
> >
> > > they definitely have some prior knowledge of the task as the commercial pay-per-query prediction NLP systems
> >
> > Sure, they know the input and output types. But assuming knowledge to the domain of the original training data seems like a bit of a stretch, the work will be more compelling if you perform the attacks with misaligned domains. Nevertheless, I appreciate the n-gram statistics and am glad to hear that you ensure there's no overlap with victim data.
> >
> > > We never assume that the attacker can access the private attributes in half of the victim data.
> >
> > I got this, but didn't write it well in my review. My concern was assuming access to a publicly available corpus (D_a) which has an identical *distribution* as the victim split (D_v), since you said you randomly split the data. While I agree there is no exact overlap, D_a and D_v are likely to share many similar properties.
> >
> > > What happens when you fine-tune it on D_a?
> >
> > I strongly recommend running a baseline *with* fine-tuning on BERT (the original pretrained model) on D_a. This will significantly strengthen this section. You are trying to check the hypothesis "the extracted model has private attribute information". The main confounding factor is knowledge which can be learnt from D_a itself. One way to disentangle these effects is to run a baseline model where you finetune BERT on D_a, checking how much of D_v can be inferred without the extracted model. Only if this baseline performs worse than your proposed attack can you conclude white-box access is helping. As an upper-bound, you could also run the same attack but on the victim model itself.
> >
> > > From these studies, we can see that it is hard to achieve a 100% transfer rate due to the general modification budget.
> >
> > Thanks for clarifying. As a follow-up, what's the modification budget you use for your perturbations? Some qualitative examples in the Appendix would be great.
> >
> > > However, following your suggestions, we did additional experiments on a variant of adv-bert, where the tokens are selected randomly on table 5
> >
> > Thanks again this is very useful! I'm a bit confused though, why would the random adv-bert attacks change so significantly from 1x to 5x, the amount of training data used for the extracted model? I guess random adv-bert is essentially a black-box attack? In fact, why is this variation present in all the black-box attacks discussed? Does the 1x / 5x mean something else?
> >
> > > There is no reason and motivation for the victim API model to generate the adversarial examples
> >
> > Agreed! What I meant was, this is one way to check if your extracted model is indeed close to the victim model. This can serve as an upperbound to strengthen the experimental results.

---

> > > ### Author Response · Authors · 2020-11-22
> > > **More experiments**
> > >
> > > Thanks for the further feedback. Here is our response:
> > >
> > > [Re: Domain mismatch]
> > > In Table 2, we have shown that albeit the domain mismatch, i.e. reviews data for AG news, and news data for Yelp, Blog, and TP-US, the extracted model still achieves comparable results (c.f. the accuracy of victim models). Table 3 shows that the genre of Blog is distant from both reviews data and news data. Nevertheless, according to AIA results in Table 4 and AET results in Table 5, our proposed attacks are still effective even when there is a domain mismatch.
> > >
> > >
> > >
> > > [Re: AIA on victim model]
> > > For AIA, it is different from AET and there is no white-box setting. However, we follow your suggestion: we first fine-tune BERT on D_a, then inferring attribute on D_v. Here are the results:
> > >
> > > |                         | AG news | Blog  | TP-US |
> > >
> > > |-------------------------|---------|-------|-------|
> > >
> > > | Majority class  | 49.94   | 49.57 | 38.15 |
> > >
> > > | Data Different (review) |         |       |       |
> > >
> > > |            1x           | 17.93   | 34.34 | 35.97 |
> > >
> > > |            5x           | 18.31   | 34.45 | 36.82 |
> > >
> > > |  Data Different (news)  |         |       |       |
> > >
> > > |            1x           | 15.76   | 33.88 | 36.92 |
> > >
> > > |            5x           | 17.91   | 35.39 | 37.68 |
> > >
> > > |        D_a -> D_v | 18.61   | 34.78 | 37.70 |
> > >
> > >
> > > From the above table (higher value means better empirical privacy, i.e., lower attack success), we can claim that conducting AIA on the extracted model (imitating white-box of the victim model) indeed is superior to the BERT fine-tuned on D_a itself.
> > >
> > > [Re: budget]
> > > As shown in adv-bert (Sun et al,), since the budget of 6 gives the best attack performance, we also use this budget for our experiments.
> > >
> > > Here are some examples:
> > > Original:
> > > mps back putin plan for regions . the russian duma backs president putin ' s plan to replace elected regional bosses with his own appointees .
> > > Adv:
> > > mps back putin plan for regions . the russian duma backs president putin ' s plan to replace elected regional boss3s with his own appointees
> > >
> > > Original:
> > > great ! think you really read these comments and critics also ; ) just keep doing that like now ! stop start you are going up to 10 !
> > > Adv:
> > > graet ! think you really read these comments and critics also ; ) just keep doing that like now ! stop start you are going up to 10 !
> > >
> > > Original:
> > > did you find out what is going on
> > > Adv:
> > > did ytou find out what is going on .
> > >
> > >
> > > We will put more examples in the Appendix of our next revision.
> > >
> > > [Re: the impact of 1x/5x ]
> > > Your understanding is correct. As stated in Section 4.2, 1x/5x are the size of queries for the extracted model. As we discussed in Section 4.4, we attribute this boost from 1x to 5x to higher fidelity to the victim model. From the perspective of interpretability, a rational model should rely on the salient words, when assigning a label. If two models function similarly, the rationale should be highly overlapped. Therefore, If the prediction of the extracted model is close to that of the victim model, an adversarial example cracking the extracted model is supposed to be easily transferred to the victim one. Hope this answers your questions.
> > >
> > > [Re: AET upper bound]
> > > If our understanding is correct, you want us to compare the accuracy of the victim on both adversarial examples and the transferred one from the extracted model. If so, here it is:
> > >
> > > |                                   | AG news | Yelp | TP-US | Blog |
> > >
> > > |---------------------------|-----------|------|-------|------|
> > >
> > > | w/o adv                      | 93.6    | 97.1 | 86.6  | 95.5 |
> > >
> > > | w/ adv                         |            |          |          |         |
> > >
> > > |           direct adv         | 78.0    | 84.8 | 69.4  | 78.5 |
> > >
> > > | transferred adv (1x) | 88.7    | 96.8 | 84.2  | 82.0 |
> > >
> > > | transferred adv (5x) | 87.5    | 95.9 | 84.5  | 79.2 |
> > >
> > > As shown in the table above, depending on the datasets, the transferred adversarial examples sometimes can achieve comparable performance, when compared to the direct adversarial attack on the victim model.

---

> > > > ### Comment · AnonReviewer2 · 2020-11-25
> > > > **Thanks for the extra experiments, but I'm still not convinced**
> > > >
> > > > > Additional AIA baselines
> > > >
> > > > I really appreciate you running this baseline which helps disentangle the effects of information present in D_a and information from the extracted model. The results show that a large amount of information can be inferred from D_a itself, and extracted models help a little bit. I also re-read your paper's section on these attacks. Unfortunately, I am still not very convinced. Firstly, I think assuming access to a public D_a (of identical distribution as D_v) is too strong an assumption. Your experiments on D_a ---> D_v make it clear that the two distributions are quite similar. Secondly, it's really weird to me that models with better extraction (like All Same, 5x) do worse on author inference (like you said, it's counterintuitive), and I don't understand how the explanation related to posterior sharpness results in this trend. Thirdly, improvements from D_a ---> D_v are small. This makes it important to run the upper bound attack success experiment I mentioned earlier (infer D_v from victim model itself). Does that get you much worse privacy? Like I'm trying to understand the margin of improvement towards the upper bound.
> > > >
> > > > > examples
> > > >
> > > > great examples! no complaints here.
> > > >
> > > > > the impact of 1x/5x
> > > >
> > > > I'm quite confused now and I don't think your reply really answered my original confusion. Aren't these attacks black-box with respect to the victim model? The attack performance should not depend on the fidelity of the extracted model. If the black-box attack changes with fidelity, it means you used the extracted model somehow to perform the attack. My best understanding is you did black-box attacks on the extracted model and then checked its success on the victim model. How is this a valid black-box baseline testing your hypothesis "extraction is necessary"?
> > > >
> > > > > AET upper bound
> > > >
> > > > No, I would have liked you to design white-box adversarial examples on the victim model, and check if they match the white-box adversarial examples of the extracted model.

---

> > > > > ### Author Response · Authors · 2020-11-25
> > > > > **further clarification and experiments**
> > > > >
> > > > > Thanks for the clarification
> > > > >
> > > > > [Re: AIA]
> > > > > Thanks for your time. Regarding the sharpness of the posterior probability, we conjecture that this counterintuitiveness can be attributed to the knowledge distillation, which is able to transfer certain dark knowledge from the victim model to the imitated model. Our concurrent work has shown that the extracted model can even benefit from a perturbed posterior probability. We also include the result of AIA on the victim model in the following table. Surprisingly,  the extracted model makes the training data even more vulnerable than the victim one on both AG news and TP-US. However, on the Blog dataset, launching AIA on the victim model indeed leaks the most information. So we conclude that AIA is data-dependent, attacking the victim model may not always give an upper bound, this phenomenon also raises significant awareness of privacy issues when deploying BERT-based APIs. Again, we hypothesise that the vulnerability is exacerbated by knowledge distillation ( up to 4% more private information leakage), where the extracted model can acquire certain dark knowledge. We will conduct an in-depth study on this in our future work.
> > > > >
> > > > > | | AG news | Blog | TP-US |
> > > > >
> > > > > |-------------------------|---------|-------|-------|
> > > > >
> > > > > | Majority class | 49.94 | 49.57 | 38.15 |
> > > > >
> > > > > | Data Different (review) | | | |
> > > > >
> > > > > | 1x | 17.93 | 34.34 | 35.97 |
> > > > >
> > > > > | 5x | 18.31 | 34.45 | 36.82 |
> > > > >
> > > > > | Data Different (news) | | | |
> > > > >
> > > > > | 1x | 15.76 | 33.88 | 36.92 |
> > > > >
> > > > > | 5x | 17.91 | 35.39 | 37.68 |
> > > > >
> > > > > | D_a -> D_v | 18.61 | 34.78 | 37.70 |
> > > > >
> > > > > |AIA on victim model      |    22.00   |    33.69  |   37.25
> > > > >
> > > > > Regarding the assumption on the same distribution, we recently realized a most recent work [1] that also investigated attribute inference and assumed that the public data follows the same distribution as the original training data, as the attributes in the different dataset can be different, it would be quite challenging to train attack models on data from different distributions. Moreover, in their work, they utilised a very simple algorithm -- relying on the relationship between attributes, which is less effective (10-20%) than the contextualised vector used by us. Nevertheless, in our work, we focus more on the subsequent attacks and demonstrate the risk of directly deploying BERT-based APIs. We agree that studying on a different distribution is also very important, and we are investigating this issue in another concurrent work.
> > > > >
> > > > > [Re: adv on black-box setting]
> > > > > Sorry for the confusion. First of all, we want to stress that as we mentioned in our paper, the black-box adversarial attack is first conducted on the extracted model, then transferred to the victim model, which is the gist of this work. Our work aims to study the subsequent potential risks after the model extraction. Hence all attacks studied in this work except the model extraction leverage the local replica as a bridge to the victim model. As shown in our paper, since black-box attacks target a particular model,  it is quite brittle. If the victim model is updated, malicious users have to launch another round of the black-box attacks. Furthermore, a black-box attack requires intensive similar queries [2, 3], which can be detected and thereafter banned by APIs hosts. Instead, the model extraction aims at functional fidelity and acts as regular users. Malicious users can run any attacks on the extracted model as many as they can, then transfer to the victim model, which has been substantiated by [4] as well.
> > > > >
> > > > > [Re: comparison between victim model and the extracted model]
> > > > > Thanks for the clarification. Here are the percentage of matched adversarial examples
> > > > >
> > > > >                            AG news       |      Yelp         |   TP-US    |    Blog
> > > > >
> > > > >       1x           |        36.54%      |      30.89%   |    43.60%  |    61.22%
> > > > >
> > > > >       5x           |        40.38%      |      34.18%   |    43.60%   |    67.65%
> > > > >
> > > > > According to this table and table 5, there is a positive correlation between the transferability and matched adversarial examples, i.e. a higher percentage of matched adversarial examples suggests a better transferability.
> > > > >
> > > > > [1] Song, Congzheng, and Ananth Raghunathan. "Information Leakage in Embedding Models." arXiv preprint arXiv:2004.00053 (2020).
> > > > >
> > > > > [2] Li, Jinfeng, et al. "Textbugger: Generating adversarial text against real-world applications." arXiv preprint arXiv:1812.05271 (2018).
> > > > >
> > > > > [3] Jin, D., Jin, Z., Zhou, J.T. and Szolovits, P., 2020, April. Is bert really robust? a strong baseline for natural language attack on text classification and entailment. In Proceedings of the AAAI Conference on Artificial Intelligence (Vol. 34, No. 05, pp. 8018-8025).
> > > > >
> > > > > [4] Wallace, Eric, Mitchell Stern, and Dawn Song. "Imitation Attacks and Defenses for Black-box Machine Translation Systems." arXiv preprint arXiv:2004.15015 (2020).

---

### Official Review · AnonReviewer1 · 2020-10-28
**Need some clarifications.**

**Rating:** 6
**Confidence:** 3

**Review:**

The paper is motivated by a challenging problem in deploying a neural network-based model for sensitive domain and research in this direction is essential for making such model usable for sensitive domains. The paper presents a model extraction attack, where the adversary can steal a BERT- based API (i.e. the victim model), without knowing the victim model’s architecture, parameters or the training data distribution. The model extraction attack, where the adversary queries the target model with the goal to steal it and turn it into a white-box model. They demonstrated using simulated experiments that how the extracted model can be exploited to develop effective attribute inference attack to expose sensitive information of the training data. They claimed that the extracted model can lead to highly transferable adversarial attacks against the original model (victim model).

The model extraction step of the proposed method is the main concern for me. Conclusions maid by simulated experiments on model extraction attack might not hold for a real experiment. The simulated experiments make both victim model and extracted model accessible and thus measuring functional similarity is fairly easy. However, without the knowledge of the victim model and with limited query budget, the simulated experiment might not resemble a real-scenario. Some explanations with real scenarios would make the claim more realistic.


Some thoughts:

Re: “Modern NLP systems are typically based on a pre-trained BERT. ”: provide references or evidence to support the statement.

Re: “Model extraction attack aims to steal an intellectual model from cloud services.”: provide references or evidence to support the statement.

Re: “Most existing adversarial attacks on BERT are white-box settings, requiring knowledge of either the model internals (e.g., model architecture, hyperparameters) or training data.”:  provide references or evidence to support the statement.

Re: “The intuition lies in the fact that the similarity of our extracted model and the victim model allows for direct transfer of adversarial examples obtained via gradient-based attacks.”  — BERT part is same for both victim and extracted model but rest is still unknown and how the complexity of the similarity measurement increases for a real scenario?

Re: “We measure the accuracy (on the same held-out test set for evaluation purposes) between the outputs of the victim model and the extracted model to assess their functional similarity.” — Can this be arbitrarily true by accident? Is there a robust way that we can use to measure the similarity?

---

> ### Author Response · Authors · 2020-11-19
> **Response to AnonReviewer1**
>
> First of all, We would like to appreciate the reviewer’s suggestion.
>
> [Re: realisticity] We would like to clarify/correct a few key points in the reviewer's comments. Our primary claim is that in spite of the achievements made by BERT-based models, there are lots of security issues when companies directly deploy them as APIs. This work raises three issues: 1) model extraction, 2) privacy leakage and 3) pseudo adversarial attack. Our paper has shown that BERT-based models are vulnerable to all these attacks. Due to the space constraint, in the main content, we only described a simpler case, where both victim and imitated model use BERT base. However, we want to stress that the BERT part is not always the same for both victim and extracted model in our paper. In Appendix Section D (c.f. original version Table 8) or Section 4.5 (c.f. revised version Table 6), we did show that without the prior knowledge of the architecture of the victim model, our attacks are still effective. We also provided limited query budgets in Appendix Table 7 by varying the number of queries from 0.1X-5X. Thus the assumption made in this paper is relatively practical, universal instead of hypothetical. Given the success of the proposed attacks, our next step is to explore a more realistic setting, i.e. how to fulfill the comparable performance to the current setting, with limited query budgets.
>
> [Re: complexity for a real scenario] We admit that the complexity of the similarity measurement between two models is intractable, especially for those that are distant from each other. However, we hold an assumption that for a given sentence, a rational model should rely on the salient words, when assigning a label. Hence, if two models function similarly, the rationale should be highly overlapped. The black-box adversarial attack leverages the brittleness of the deep learning model, whereas the white-box attack seeks the most informative words affecting the decision according to the gradient [1]. According to our assumption, the informative words should be easily transferred to the victim, which is also verified by Table 5 and Table 6 in the revised version. Of course, this assumption requires further rigorous study.
>
>
> [thoughts1-5] For model extraction attack, we have clearly cited [2, 3, 4, 5] in Sec.2.1. We added these references to the statement mentioned by the reviewer to make it more clear.
> We have modified the statement “Most existing adversarial attacks on BERT are white-box settings...” to the following: However, most recent works for adversarial example transfer focus on the black-box setting [6, 7]. In such a setting, the adversary attacks the model via the query feedback only in Sec.3.4.
> In Table 3, Table 8 (c.f. Appendix Section D in the original version) or Table 6 in our revision, Table 7 (in Appendix), we have shown how the complexity of the similarity measurement increases for more realistic scenarios: when the attacker has no prior knowledge of the architecture of the victim model, no prior knowledge of the victim model training data distribution, and only has limited query budget.
> Definitely there should be a robust way that we can use to measure the similarity, which however is out of the scope of this paper. Our main focus is to exploit the privacy leakage and vulnerability of transferred adversarial examples from the extracted model.
>
> [1] Sun, Lichao, et al. "Adv-BERT: BERT is not robust on misspellings! Generating nature adversarial samples on BERT." arXiv preprint arXiv:2003.04985 (2020).
>
> [2] Kalpesh Krishna, Gaurav Singh Tomar, Ankur P Parikh, Nicolas Papernot, and Mohit Iyyer. Thieves on sesame street! model extraction of bert-based apis.arXiv preprint arXiv:1910.12366, 2019
>
> [3]  Knockoff nets:  Stealing functionality of black-box models.  InProceedings of the IEEE Conference on Computer Vision and Pattern Recognition, pp. 4954–4963, 2019
>
> [4] Stealing machine learning models via prediction apis. In25th{USENIX}Security Symposium ({USENIX}Security16), pp. 601–618, 2016
>
> [5] Eric Wallace, Mitchell Stern, and Dawn Song. Imitation attacks and defenses for black-box machine translation systems.arXiv preprint arXiv:2004.15015, 2020
>
> [6] Ji Gao, Jack Lanchantin, Mary Lou Soffa, and Yanjun Qi. Black-box generation of adversarial text sequences to evade deep learning classifiers. In2018 IEEE Security and Privacy Workshops (SPW),pp. 50–56. IEEE, 2018.
>
> [7] Javid Ebrahimi, Daniel Lowd, and Dejing Dou. On adversarial examples for character-level neural machine translation.arXiv preprint arXiv:1806.09030, 2018a

---

### Official Review · AnonReviewer3 · 2020-10-29
**Expanded Discussion of Impact and Limitations**

**Rating:** 4
**Confidence:** 3

**Review:**

################################

Summary:

This paper presents a model extraction attack (MEA) for BERT-based models that are hosted behind an API. Using the model obtained in this step, the work aims to subsequently demonstrate attribute inference attacks (AIA) to expose sensitive information of the underlying data used during fine-tuning and adversarial example transfer (AET) that can be used to attack the hosted model.

################################

Reasons for score:

Overall, I lean toward reject. The underlying idea is interesting and timely, and core to this interest is that "the adversary can steal a BERT-based API (the victim model), without knowing the victim model's architecture, parameters or the training data distribution." As demonstrated, a substantial portion of the architecture (BERT) is known and the exploration of fine-tuning as the only mechanism for tailoring the model (rather than continual pretraining) limits potential impact.

################################

Strengths:

- Broad interest. The underlying ideas are of general interest, especially given recent examples of language models hosted behind APIs. The notion that they can be efficiently reproduced from that API and that they may in turn leak training information is an emerging concern.

- Clear differentiation from prior work. In particular, the section on comparison to knowledge distillation is helpful in grounding the setting for experimentation.

################################

Weaknesses:

- Limitations. There is an implicit assumption that the models being hosted behind APIs are fine-tuned BERT models. Limitations of this should be more explicitly discussed. Many works in specific domains (e.g., legal, biomedical, etc.) appear to rely on continual pretraining to integrate sensitive data rather than fine-tuning toward a single task. Others even appear to train these models from scratch on data. It's unclear how common the case of fine-tuned BERT models behind APIs are from this paper.

- Motivation of AET. The motivation of adversarial attacks against a pay-per-query API are unclear. Yes, it's possible to cause the API to create incorrect predictions, but why is that problematic for the owner of the model? It's clearly undesirable with respect to creating robust models, but as presented it's unclear why this is problematic.

- Impact. Similar to the point above, the assertion that "modern NLP systems typically leverage the fine-tuning methodology by adding a few task-specific layers on top of the publicly available BERT base" is not substantiated by this work or by citation. While BERT has certainly become abundant, many recent advances are either not BERT-based (though perhaps the underlying transformer architecture) or do more than fine-tuning.

- Knowledge of black-box model. While a stated goal is that a knowledge of the architecture and training data is not required, the experiments leverage a knowledge of the architecture (BERT) and appear to share an architecture for layers used during fine-tuning.

################################

Questions:

- Given the positioning of "stealing" a model, how many queries are required to obtain an approximate model? How many are required if knowledge of the previously issued queries is known?

- Can you provide pointers to models that are BERT-based and fine-tuned for a specific task only?

---

> ### Author Response · Authors · 2020-11-19
> **Response to AnonReviewer3**
>
> First of all, We would like to appreciate the reviewer’s suggestion.
>
> [Re: Limitation] We agree with the weakness pointed by the reviewer. However, we would like to highlight our main contribution: we aim to raise general issues related to BERT models, it is not restricted to the original BERT models, but can also be applied to any domain-specific models (SciBERT (Beltagy et al. 2019), FinBERT (Yang et al. 2020), BioBERT (Lee et al. 2019), and so on). Since all these models are publicly available, the attacker can easily adapt them for the task of interest under the same attacking protocol introduced in this work.
>
> Additionally, domain-specific unlabelled datasets can be crawled easily as well. As we described in Table 2, model extraction can also be beneficial to different datasets at the cost of more queries. Krishna et al. 2019 have shown that using even dummy queries can steal fine-tuned task-specific BERT models with competitive performance.
>
> Finally, we experimented with different architectures in Appendix Section D (c.f. original version Table 8) or Section 4.5 (c.f. revised version Table 6). As shown in these tables, even though the victim models are different from the extracted model, the proposed attacks are still effective, which draws an analogy to knowledge distillation, where the teacher model and the student model can be different.
>
> To sum up, the proposed attacks are model-agnostic. One can plug and play any publicly available pretrained models and dataset for a particular task.
>
> [Re: Motivation of AET] Malicious users could be one of the business competitors. Hence if they manage to spot incorrect predictions, they can improve the robustness of their model while launching an advertising campaign against the victim model with these adversarial examples. If a rival company directly studies adversarial attacks on the victim model, its owner can spot the suspicious querying and ban it. We have made it clear in our revision. For the motivation of AET, it’s also recommended to refer to the most recent work [Wallace et al. 2020], which transfers adversarial examples to machine translation systems.
> Lastly, a malicious user can utilise misclassification or mistranslation to raise any controversial quarrel or even disturbance, if there are political, ethical, gender, or race biases being involved. We have experienced such situations over the past years.
>
> [Re: Impact and Knowledge of black-box model]  We would like to emphasize that we did experiment with different architectures in Appendix Section D (c.f. original version Table 8) or Section 4.5 (c.f. revised version Table 6). As shown in these tables, albeit the performance degradation of all attacks,  the architectural difference between the victim and the extracted model cannot prevent the attacks launched by malicious users. Although we only experimented with BERT, RoBERTa, and XLNET, we believe the proposed scenario is a practical concern adversely affecting and being applicable to all cloud APIs. We justify this belief with the following reasons: To the best of our knowledge, despite the different training objectives (ELECTRA v.s BERT), the amount of the training data and training time (RoBERTa v.s. BERT), and so on, all the publicly available pre-trained models share the same architecture, i.e. transformer. Hence experiments on BERT, RoBERTa, and XLNET are sufficiently representative.
>
> [Re: Questions] As shown in Table 2, Table 7 (in Appendix) and all previous works (Krishna et al 2019, Orekondy et al. 2019), the number of queries to obtain an approximate model is varied across different tasks, and generally more queries result in better extraction. The study on the size of queries is also a very interesting research direction, which would be our future work and we did not take it as the main focus of this work. We want to emphasize that this work pays more attention to the subsequent attacks based on the extracted model, which is only served as a prerequisite, but not the gist of this work.
>
> Here are a list of papers fine-tuning on BERT:
> SemEval-2019 Task 6: Identifying and Categorizing Offensive Language in Social Media (OffensEval)(https://arxiv.org/pdf/1903.08983.pdf)
>
> Passage Re-ranking with BERT (https://arxiv.org/abs/1901.04085)
>
> Text Summarization with Pretrained Encoders (https://arxiv.org/abs/1908.08345)
>
> SpanBERT: Improving Pre-training by Representing and Predicting Spans (https://arxiv.org/abs/1907.10529)
>
> In addition, according to huggingface website, bert-base-uncase is the most downloaded model among all the pretrained model (https://huggingface.co/models)
>
> Although recent works have modified the objective or architecture to fit their tasks, they are still derived from BERT. Our work leverages BERT-based as a case study, but both the methodology and security issues are universal and model-agnostic. Other works can alter the proposed attacks for a particular scenario.

---

### Official Review · AnonReviewer4 · 2020-10-30
**Reviews and Comments**

**Rating:** 6
**Confidence:** 4

**Review:**

### Overview

The authors propose a pipeline to attack and steal sensitive information from a BERT-based API service, and can subsequently perform adversarial attack to the victim model by creating white-box adversarial samples on the stolen model.

The pipeline can be summarized as the followings:
1. Using distillation to train (steal) a model from the API.
2. Conduct model inversion attack to the stolen model in step 1 to expose sensitive information of the training data.
3. Create adversarial samples for the stolen model in step 1 and use them to attack the original API.

Some of the assumptions of the experiment settings are too strong and far from the real situation, but the idea of using this pipeline to conduct model inversion and adversarial transfer attack is very interesting.

### Pros

The pipeline proposed by the authors is very insightful. The experiment results also show the effectiveness of model inversion and adversarial attack.

### Cons

The assumption of the Model Extraction Attack part is too strong. The authors use the same pre-trained BERT parameters for both victim and stealer model. However in real practice we are not able to know which pre-trained BERT parameter set is to be used for fine-tuning, nevertheless to get the pre-trained model. What if we use different pre-trained BERT parameters? What if we use a pre-trained BERT with different size (num of layers, hidden dim, etc.)?

Besides, the stealing method is just a conventional distillation. Although the authors claims three differences between their method and distillation: (1) the goal (2) the accessibility of original data (3) the accessibility of hard labels, only the first one is appropriately claimed. For (2), distillation is also broadly used in transfer learning w/o the access of original data. For (3), distillation w/ only soft labels is also very popular and useful, from conventional distillation for compression, to self-distillation.

I'd like to hear the reason why the authors make the assumption that the stealer would have the same pre-trained BERT when attacking, and also curious about the results of using different pre-trained BERT model. I might change the rating if the authors may address these questions.

---

> ### Author Response · Authors · 2020-11-19
> **Model extraction with different architectures**
>
> We would like to appreciate the reviewer’s suggestion. Although in the main content, we hold a strong assumption, we also provided more experimental results on more realistic scenarios, where the victim and imitated model have a different architecture, see Appendix Section D (c.f. original version Table 8) or Section 4.5 (c.f. revised version Table 6). As shown in these tables, albeit the performance degradation of all attacks,  the architectural difference between the victim and the extracted model cannot prevent the attacks launched by the attacker. Although we only experimented with BERT, RoBERTa, and XLNET, we believe the proposed scenario is a practical concern that may adversely affect all cloud APIs. We justify this belief with the following reasons: To the best of our knowledge, despite the different training objectives (ELECTRA v.s BERT), the amount of the training data and training time (RoBERTa v.s. BERT), and so on, all the publicly available pre-trained models share the same architecture, i.e. transformer. Hence experiments on BERT, RoBERTa, and XLNET are sufficiently representative.
>
> Moreover, the gist of our work is not about knowledge distillation (KD). Instead, KD is merely a vehicle that bridges the victim model and the extracted model. Our key contribution is to leverage a surrogate to conduct successive attacks. If one directly studies some black-box attacks on cloud APIs, the owner of the victim model can spot the intensive abnormal querying actions and ban any queries from the attackers. In our revision, we have deleted the comparison to KD to avoid any confusion.

---

> > ### Comment · AnonReviewer4 · 2020-11-25
> > **Reply**
> >
> > I just realized I didn't notice the pointer to Appendix D about the discussion of different architecture pairs in the original version. It's much better now as the authors put it to the main body. I believe it is a common concern of a large audience from the production side.
> >
> > About distillation, I'm not criticizing the contribution. Given the pipeline proposed in the paper is already novel and effective, I would suggest the authors not emphasizing too much on the difference of distillation methods. As I mentioned, the three differences claimed is not very appropriate. In my personal opinion, KD is already a well-known technique in the machine learning community and you may just mention it briefly and leave some pointers to some comprehensive surveys.
> >
> > I'd like to change my rating to 6; or, AC can treat it as a rating of 5.5~6 in a strictest view.

---

### Decision · Program_Chairs · 2021-01-07
**Final Decision**

**Decision:**

Reject

**Comment:**

The paper presents novel model stealing attacks against BERT API. The attacks are split in two phases. In the first phase, the black-box BERT model is recovered by submission of specially crafted data. In the second phase, the inferred model can be used for identifying sensitive attributes or to generate adversarial examples against the basic BERT model.

Despite the novelty of presented attacks against BERT models, the current version of the paper has some problems with clarity and motivation. The presentation of attacks is very short, and some technical details are not adequately covered. The practical motivation of adversarial example transfer attacks is not very clear, and the authors' response on this issue did not provide a convincing clarifications. Furthermore, creation of surrogate models for generation of adversarial examples is a well-known technique and the difference of the proposed AET attack from this conceptual approach is not clear.

Overall, the paper reveals a solid and interesting work but a substantial revision would be necessary to make it suitable for the ACLR audience.